# UVRAG Promotes Tumor Progression through Regulating SP1 in Colorectal Cancer

**DOI:** 10.3390/cancers15092502

**Published:** 2023-04-27

**Authors:** Mengyuan Shi, Guo An, Nan Chen, Jinying Jia, Xinxin Cui, Tiancheng Zhan, Dengbo Ji

**Affiliations:** 1Key Laboratory of Carcinogenesis and Translational Research (Ministry of Education), Department of Gastrointestinal Surgery III, Peking University Cancer Hospital & Institute, Beijing 100142, China; 2Key Laboratory of Carcinogenesis and Translational Research (Ministry of Education/Beijing), Department of Laboratory Animal, Peking University Cancer Hospital & Institute, Beijing 100142, China

**Keywords:** UVRAG, prognosis, microenvironment, tumor malignancy, colorectal cancer

## Abstract

**Simple Summary:**

UVRAG has been identified as being involved in tumor progression and prognosis in various types of cancer. We aimed to find the relationship between UVRAG and the prognosis of colorectal cancer and its potential mechanisms. We found that UVRAG had a negative association with prognosis in colorectal cancer, which was mediated by its ability to enhance migration, stemness, and chemoradiotherapy resistance of cancer cells. Additionally, UVRAG can promote the recruitment of macrophages by upregulating SP1 and enhance PD-L1 expression. This study deepens the understanding of UVRAG from the aspect of tumor immunity and provides more evidence for UVRAG as a novel therapeutic target with or without being combined with immunotherapy or molecular inhibitors.

**Abstract:**

Colorectal cancer (CRC) is the third most common type of cancer. The ultraviolet radiation resistance-associated gene (UVRAG) plays a role in autophagy and has been implicated in tumor progression and prognosis. However, the role of UVRAG expression in CRC has remained elusive. In this study, the prognosis was analyzed via immunohistochemistry, and the genetic changes were compared between the high UVRAG expression group and the low UVRAG expression group using RNA sequencing (RNA-seq) and single-cell RNA-seq (scRNA-seq) data, and genetic changes were then identified by in vitro experiments. It was found that UVRAG could enhance tumor migration, drug resistance, and CC motif chemokine ligand 2 (CCL2) expression to recruit macrophages by upregulating SP1 expression, resulting in poor prognosis of CRC patients. In addition, UVRAG could upregulate the expression of programmed death-ligand 1 (PD-L1). In summary, the relationship between UVRAG expression and the prognosis of CRC patients as well as the potential mechanisms in CRC were explored, providing evidence for the treatment of CRC.

## 1. Introduction

Colorectal cancer (CRC) is one of the most common types of cancer worldwide. The advent of radical surgery and 5-fluorouracil (5-Fu)-based adjuvant treatment has significantly improved survival rates [1]. The ultraviolet radiation resistance-associated gene (UVRAG), located in 11q13, encodes a 63-kDa cytoplasmic protein, which has been reported to be involved in the processes of autophagy and apoptosis [2,3,4]. UVRAG promotes autophagy by regulating Beclin 1-PI3KC3 complex as a positive mediator [5] and suppresses Bax-induced apoptosis by influencing the location of Bax [6]. Moreover, UVRAG has been found to maintain chromosomal stability by repairing DNA double-strand breaks (DSBs) and stabilizing centrosomes [7]. UVRAG is regarded as a suppressor gene in diverse types of cancer due to its function of accelerating autophagy [8,9]. However, evidence has shown that increased autophagy is mainly present in cancer cells to facilitate their survival in different environments [10]. Sevide Sencan et al. demonstrated that UVRAG promotes the proliferation, migration, and invasion of breast cancer cells [11]. While UVRAG has been studied in multiple types of cancer, its function in CRC has not yet been fully clarified. The present study aimed to explore the relationship between UVRAG and CRC prognosis, treatment response, and the underlying mechanisms.

## 2. Methods

### 2.1. Patients and Samples

The study was approved by the Ethics Committee of Peking University Cancer Hospital & Institute (Beijing, China), and it was performed in accordance with the Declaration of Helsinki. All patients were informed about the objectives of the study and signed the informed consent form before enrollment. For Cohort 1, 294 adult patients who met the following criteria were included: (1) age ≥ 18 years old, (2) diagnosis with CRC, (3) patients who did not receive neoadjuvant treatment, and (4) undergoing surgery in Beijing Cancer Hospital from 1999 to 2006. For Cohort 2, 296 adult patients with CRC were enrolled, and they received neoadjuvant radiotherapy of a 30Gy dose in 10 fractions for 2 weeks, followed by surgery. Tissue microarrays (TMAs) generated by surgical samples of patients from Cohort 1 and Cohort 2 were subsequently used for immunohistochemistry (IHC) analysis. Cohort 3 included 60 patients whose preneoadjuvant chemoradiotherapy (pre-nCRT) biopsies were obtained from Cohort 2, and these fresh samples were analyzed by quantitative polymerase chain reaction (qPCR). The detailed characteristics of the enrolled patients are presented in Appendix A.

### 2.2. Assessment of Tumor Downstaging

The efficacy of neoadjuvant radiotherapy was assessed via the tumor regression grade (TRG) system. The following criteria were used to grade TRG [12]: Grade 0, absence of regression; Grade 1, major residual tumor with obvious fibrosis; Grade 2, preliminary fibrotic changes with residual tumor; Grade 3 (approximately pathological complete response (pCR)), rare residual tumor cells in fibrotic tissue; and Grade 4, complete response (CR) with no tumor cells.

### 2.3. Cell Lines and Cell Culture

Human CRC cell lines (HCT116, HT29, and HCT8) were obtained from the American Type Culture Collection (ATCC; Manassas, VA, USA) and cultured in a Roswell Park Memorial Institute (RPMI)-1640 medium (Gibco, Carlsbad, CA, USA) containing 10% fetal bovine serum (FBS) in a humidified incubator at 37 °C in the presence of 5% CO_2._ These cell lines were tested and identified by short tandem repeat analysis. Cells were routinely tested for mycoplasma infection and could be used only when the test result was negative.

HCT8-5-Fu refers to HCT8 cell lines with resistance to chemotherapy, which is established by exposure to the increased 5-Fu concentration. HCT8-60Gy is defined as radio-resistant HCT8 generated by exposure to 12 × 5 Gy X-ray, followed by recovery for 4 weeks. HCT116-self is defined as stem cell-enriched cell spheres established by a previously reported method [13].

### 2.4. TMAs and IHC

TMAs in Cohort 1 included 294 primary tumor tissues and 72 paired adjacent normal colorectal mucosa and 32 matched metastatic liver tissues. Four points were considered for each primary tumor and liver metastasis, and 2 points were considered for each normal tissue by a 0.4-mm needle, and they were then arranged on slides. For Cohort 2, TMAs were established by tissues from 296 primary tumors and their corresponding normal tissues, with 4 points for each sample. In addition, IHC staining of UVRAG (UVRAG (D2Q1Z) Rabbit mAb, #13115; Cell Signaling Technology, Danvers, MA, USA), programmed cell death protein 1 (PD-1; PD-1 LS-B540; LifeSpan BioSciences, Seattle, WA, USA), programmed death-ligand 1 (PD-L1; #ab58810; Abcam, Cambridge, UK), and PD-L2 (#HPA013411; Sigma-Aldrich, St. Louis, MO, USA) were performed. All images were examined by two experienced pathologists independently. Samples were analyzed with 2 or more tissue points. UVRAG was assumed to be located in the cytoplasm, and the results were evaluated based on the positive control from Cell Signaling Technology and negative control from slides without anti-UVRAG antibody. The expression of UVRAG was determined by assessing the intensity of staining as well as the percentage of staining cells. Specifically, tissues that showed no staining were assigned a score of 0. Tissues that showed a staining with less than 25% of cells were scored as 1. Tissues with strong staining observed in 25–50% of cells were scored as 2, while those with strong staining in more than 50% of cells were given a score of 3. The patients with low UVRAG expression are those who were scored as 0 or 1, and the patients with high UVRAG expression are those who were scored as 2 or 3.

### 2.5. Transient Transfection

The small interfering RNAs (siRNAs) targeting UVRAG were obtained from RiboBio Co., Ltd. (Guangzhou, China), and plasmids containing the target gene UVRAG were achieved from GENECHEM Co., Ltd. (Shanghai, China). HT29 cells were seeded into 6-well plates (500,000 cells per well) and were transiently transfected using riboFECT CP (RiboBio Co., Ltd.) after being cultured in a complete medium for one day. Similarly, HCT116 cells were seeded into 6-well plates (500,000 cells per well) and were then transfected using Lipofectamine 3000 reagent (Thermo Fisher Scientific, Waltham, MA, USA). The subsequent experiments were performed after transfection for 72 h. Twenty-four hour after UVRAG infection, SP1 was infected using Lipofectamine 3000 reagent and incubated for 48 h.

### 2.6. Cell Cytotoxicity Assay

Cell viability was assessed via a cell counting kit-8 (CCK-8) assay kit (Dojido, Kumamoto, Japan). Cells were seeded into 96-well plates with 10,000 cells and 100 μL RPMI-1640 medium per well and received gradient concentrations of 5-Fu for 48 h. Five duplicates were set, and cell survival rates were calculated with normalized proportions by GraphPad Prism 8.0 software (GraphPad Software Inc., San Diego, CA, USA).

### 2.7. Cell Migration Assay

Cell migration was assessed using the Boyden chamber assay. For this purpose, 1 × 10^5^ UVRAG overexpressing HCT116 cells were seeded onto the upper chamber of transwell inserts in 200 μL RPMI-1640 medium (Gibco) without FBS, and the lower chamber included 700 μL RPMI-1640 medium containing 10% FBS. After incubation for 48 h, the chambers were fixed with formalin and stained with 0.1% crystal violet, and the images were then captured using a DMi8 microscope (Leica, Wetzlar, Germany).

### 2.8. Clonogenic Survival Assay

Single-cell suspensions of UVRAG overexpressing HCT8 (HCT8-OE) and HCT8 control (HCT8-CON) cells were seeded into 96-well plates in specified numbers in triplicate for 24 h and were then treated with radiation (0, 2, 4, 6, or 8 Gy). After cell culture for 10 days in a humidified incubator at 37 °C with 5% CO_2_, the whole plates were stained with crystal violet, and survival fractions (SFs) were calculated as follows: SFs = colonies counted/cells seeded × (plating efficiency (PE)/100). The radiation enhancement ratio (ER) was calculated as follows: ER = mean inactivation dose in the experimental group/mean inactivation dose in the control group. ER significantly < 1 indicates radio resistance.

### 2.9. Spheroid Formation Assay

A total of 1200 cells were suspended in a solution consisting of 1200 µL of a 1:1 mixture of methylcellulose with EGF supplemented and a culture medium that was free of FBS. A total of 200 cells per well were seeded in ultralow-attachment 96-well plates with 6 replicates. Then, we incubated the 96-well plates in a cell culture incubator with 5% CO_2_ at 37 °C to allow spheroids to form. After 5 days, we used a microscope to assess the spheroid formation and take pictures using a DMi8 microscope (Leica, Wetzlar, Germany).

### 2.10. Western Blot Analysis

Western blotting was used to detect the expression levels of UVRAG, Sp1 transcription factor (SP1), PD-L1, and PD-L2. UVRAG knocked-down HT29 cells, and UVRAG overexpressed HCT116 cells were lysed by radioimmunoprecipitation assay (RIPA) buffer. Ten micrograms of cell lysates were loaded and separated by 8–10% sodium dodecyl-sulfate-polyacrylamide gel electrophoresis (SDS-PAGE), and the separated proteins were then transferred onto polyvinylidene difluoride (PVDF) membranes via wet-transfer mode. After that, PVDF membranes were blocked with a buffer containing 5% nonfat milk for 1 h at room temperature. Membranes were incubated with the following primary antibodies overnight at 4 °C: Rabbit polyclonal anti-Glyceraldehyde 3-phosphate dehydrogenase (GAPDH) (#AP0066; Bioworld, Irving, TX, USA) diluted by 1:15,000, Rabbit monoclonal anti-UVRAG (#13115; Cell Signaling Technology) diluted by 1:1000, Rabbit monoclonal anti-SP1 (#ab124804; Abcam) diluted by 1:1000, Rabbit monoclonal anti-PD-L1 (#ab58810; Abcam) diluted by 1:500, anti-ABCG2 (#AP1490b-ev; Abgent, San Diego, CA, USA) diluted by 1:1000, anti-CD133 (#ab222782; Abcam) diluted by 1:2000, and anti-BMI1 (#ab126783; Abcam) diluted by 1:1000. After washing with Tris-buffered saline with Tween (TBST), membranes were incubated with horseradish peroxidase (HRP)-linked secondary antibody for 1 h, and the images were then captured by Amersham Imager 680 software.

### 2.11. Quantitative Reverse Transcription-PCR (RT-qPCR)

RNA extraction and reverse transcription were initially performed. Total RNA was extracted using TRIzol Reagent (Invitrogen). cDNAs were synthesized from 2 μg total RNA using oligo(dT), primers, and dNTP mixture (TaKaRa). Subsequently, RT-qPCR was performed on a 7500 fast Real-Time PCR machine, and the following primers were used: UVRAG forward primer, 5-ATGTTTTAAGCCATTATTTA-3; UVRAG reverse primer, 5- CGTTCCAGTTCATTCTG-3; GAPDH forward primer, 5-TGCACCACCAACTGCTTAGC-3; GAPDH reverse primer, 5-GGCATGGACTGTGGTCATGAG-3; CCL2 forward primer, 5-CAGCCAGATGCAATCAATGCC-3; CCL2 reverse primer, 5-TGGAATCCTGAACCCACTTCT-3; SP1 forward primer, 5-TGGCAGCAGTACCAATGGC-3; and SP1 reverse primer, 5-CCAGGTAGTCCTGTCAGAACTT-3. A final volume of 10 µL PCR reaction system consisted of 5 µL SYBR green master mix (Toyobo Co. Ltd., Osaka, Japan), 0.6 µL primer mix, 3.4 µL ddH_2_O, and 1 µg cDNA. Three control groups were set for each sample, and the relative quantification (RQ) of UVRAG to GAPDH was evaluated by calculating 2^−ΔΔCt^ (ΔCt = Ct(target) − Ct(reference); ΔΔCt = ΔCt(experiment) − ΔCt(control(GAPDH))).

### 2.12. RNA-Seq Bulk Data Analysis

The RNA-seq bulk dataset (GSE15781) was obtained from the Gene Expression Omnibus (GEO) database and used to compare UVRAG expression before and after undergoing neoadjuvant treatment (nCRT). The Cancer Genome Atlas (TCGA) data were downloaded and used to analyze gene expression and microenvironment by R 4.1.2 software with the following packages: “TCGAbiolinks”, “ggplot2”, “DESeq2”, “enrichplot”, “clusterProfiler”, and “DMwR2”. ssGSEA (single sample Gene Set Enrichment Analysis) was conducted using the “gsva” package.

### 2.13. Single-Cell RNA-Seq (scRNA) Data Analysis

Raw data of single-cell sequencing (GSE188711) were downloaded from the GEO database and were then primarily analyzed using the “Seurat” R package. After filtering low-quality cells under the criteria of nFeature_RNA > 400, nCount_RNA > 1000, nCount_RNA < 20,000, and percent.mt < 20, 21,579, cells were retrieved for subsequent analysis. After normalization and standardization, dimension reduction was conducted by the t-distributed stochastic neighbor embedding (t-SNE) algorithm. The “FindAllMarkers” function was then used to obtain the marker genes of each cluster, which was applied for subsequent manual annotation. In addition, “SingleR” R package was utilized to confirm the manual annotation. The gene expression in different groups was estimated using log_2_(TMP + 1), the *t*-test was used for comparing two groups, and the analysis of variance (ANOVA) test was employed for comparing multiple groups. To explore the interaction among different clusters, the “Cellchat” R package was utilized to estimate the intensity of receptor–ligand pairs.

### 2.14. Statistical Analysis

Overall survival (OS) was defined as the time from the date of surgery to the date of death. Disease-free survival (DFS) was defined as the time from the date of surgery until the first recurrence or death due to any cause, whichever was observed first. The survival variables between the high UVRAG expression group and the low UVRAG expression group were analyzed by the Kaplan-Meier method, and survival curves were compared using the log-rank test. In addition, a Cox regression model was established for the multivariate analysis. The differences between two or more groups for abnormally distributed variables were assessed via the Mann–Whitney U test and the Kruskal–Wallis test, respectively. The χ^2^ test was used to compare enumeration data, and *p* < 0.05 was considered statistically significant. The statistical analysis was performed using SPSS 17.0 software (IBM, Armonk, NY, USA).

## 3. Results

### 3.1. Correlation between UVRAG and Clinicopathological Features of CRC Patients without Neoadjuvant Treatment

To investigate the potential association between UVRAG and clinical and pathological features of CRC patients, IHC analysis of Cohort 1 was performed (Figure 1A). Correlation analysis revealed that UVRAG had a positive correlation with tumor differentiation (*p* = 0.016), distant metastasis (*p* < 0.01), TNM stages (*p* = 0.008), PD-1 expression (*p* = 0.041), PD-L1 expression (*p* = 0.000), and PD-L2 expression (*p* = 0.000). On the other hand, UVRAG showed an irrelevance with the T stage, the N stage, histological type, tumor location, and vascular cancer embolus (Table 1). In addition, PD-L2 expression was positively associated with tumor location (*p* = 0.046) (Table 2). Further exploration of the association of UVRAG expression and expressions of PD-1, PD-L1, and PD-L2 in different tumor locations indicated that all of the above-mentioned markers were positively associated with UVRAG expression in colon cancer (Table 3), while only PD-L2 expression showed a correlation with UVRAG expression in rectal cancer (Table 4).

### 3.2. The Relationship between UVRAG and Long-Term Survival in CRC

The relationship between UVRAG and long-term survival in Cohort 1 was further analyzed. Survival analysis revealed that the OS rate of CRC patients in the high UVRAG expression group was significantly lower than that in the low UVRAG expression group (*p* = 0.003) (Figure 1B). For right-sided colon cancer, UVRAG expression had a significant association with poor OS (Figure 1C), while the association between UVRAG expression and the prognosis was not significant in left-sided colon cancer (Figure 1D). CD44, as a marker of cancer stem cells (CSCs) [14], was negatively associated with both DFS (*p* = 0.039) and OS (*p* = 0.012) in the present study (Figure 1E,F). In addition, it was found that patients in the low UVRAG expression group had longer survival than those in the high UVRAG expression group when patients were CD44-positive (*p* = 0.001) (Figure 1G), while the difference was insignificant in patients with CD44-negative (*p* = 0.305) (Figure 1H). Moreover, in patients with p53 mutation, those with low UVRAG expression showed a higher OS rate than those with high UVRAG expression (*p* = 0.01) (Figure 1I). However, no significant difference was found in patients with wild-type p53 (Figure 1J).

### 3.3. UVRAG Expression Was Correlated with the Efficacy of Adjuvant Chemotherapy

To clarify the relationship between UVRAG expression and the efficacy of adjuvant chemotherapy, the prognosis of CRC patients in the high UVRAG expression group and the low UVRAG expression group was compared under the condition of receiving or not receiving adjuvant chemotherapy in Cohort 1. For patients who received adjuvant chemotherapy, the high UVRAG expression was correlated with the reduced OS rate (*p* = 0.011) (Figure 2A), whereas in patients not receiving adjuvant chemotherapy, there was no significant difference in OS rates between the high and low UVRAG expression groups (Figure 2B).

### 3.4. UVRAG Expression Was Correlated with the Efficacy of Radiotherapy and Prognosis following Neoadjuvant Radiotherapy

The UVRAG expression of patients with and without a history of receiving radiotherapy was compared using RNA-seq data from 42 patients with advanced CRC in the GEO (GSE15781), and it was found that UVRAG expression was higher in patients who received radiotherapy (Figure 2C). In addition, the UVRAG expression in pre-nCRT biopsies of Cohort 3 by RT-qPCR revealed that UVRAG expression was notably higher in patients with TGR scores of G1 and G2 than in those with TGR scores of G3 and G4 (Figure 2D), indicating that the higher UVRAG expression might be associated with the worse efficacy of radiotherapy.

For Cohort 2, after receiving neoadjuvant radiotherapy, UVRAG expression was correlated with expressions of PD-L1 and PD-L2 in rectal cancer (Table 5), and patients with positive PD-L2 expression in the high UVRAG expression group exhibited a worse prognosis (Figure 2E,F).

### 3.5. UVRAG Contributed to Resistance to Radiotherapy and Chemotherapy in CRC

To clarify whether UVARG plays a role in chemotherapy and radiotherapy resistance, we compared the UVARG expression in wild-type HCT8 and chemoradio-resistant HCT8-5Fu/HCT8-60Gy. The UVRAG expression in 5-Fu-resistant HCT8 (HCT8-5Fu) and radio-resistant HCT8 (HCT8-60Gy) cells was higher than that in control cells (Figure 3A), indicating that UVARG might promote chemoradiotherapy resistance in CRC. The clonogenic survival assay was performed, and it was found that HCT8 cells with UVRAG overexpression presented an enhanced radiotherapy resistance (ER = 0.54) (Figure 3B). Compared with the control group, UVRAG knockdown increased the sensitivity of HT29 cells to 5-Fu (IC50: 20.83 μM in the control group vs. 4.5 μM in the knockdown group) (Figure 3C). Similarly, HCT116 cells with UVRAG overexpression exhibited reduced 5-Fu sensitivity (IC50: 31.44 μM in the control group vs. 66.37 μM in the overexpression group) (Figure 3D).

Stemness is a prominent property that has been linked to chemoradiotherapy resistance. To explore the connection between UVARG and stemness, we compared the wild-type HCT116 to HCT116-self. HCT116-self cells were derived from stem cell-enriched cell spheres. UVRAG expression in HCT116 cells was predominantly lower than that in HCT116-self cells (Figure 3E). Doublecortin-like kinase 1 (DCLK) was reported as a marker of CSCs, involved in resistance to 5-Fu and radiotherapy [15,16]. Therefore, UVRAG expression in DCLK+ HCT116 and DCLK- HCT116 cells was detected, and it was found that DCLK+ HCT116 cells showed a higher UVRAG expression (Figure 3F), suggesting that UVRAG might be associated with CRC stem cells. The high level of UVARG expression observed in cell lines with increased stemness suggests that UVARG contributes to chemoradiotherapy due to its ability to enhance stemness. To further investigate the correlation between stemness and UVRAG, we measured the expression of CSCs markers. We found that the expressions of CD133, ABCG2 (ATP Binding Cassette Subfamily G Member 2), and BMI-1 (B-lymphoma Mo-MLV insertion region 1) were upregulated in UVRAG overexpressed cell lines and downregulated in UVRAG knocked-down cell lines (Figure 3G). This suggests that UVRAG has the ability to amplify the stemness of cancer cells. The spheroid formation assay showed that the self-renewal ability of UVRAG overexpressed cell lines was significantly stronger than that of control cell lines, identifying the role of UVRAG in regulating CSCs properties (Figure 3H). Moreover, we used ssGSEA to deduce the proportion of stem cells in cancer tissues from patients with colon cancer and found that patients with higher levels of UVRAG expression had a higher proportion of CSCs than those with lower levels of UVRAG expression (Figure 3I).

### 3.6. UVRAG Recruited Macrophages through CCL2 by Regulating Transcriptional Factor SP1

Using RNA-seq data of 524 patients with colon cancer from TCGA, the difference in immune infiltration between the high UVRAG expression group and the low UVRAG expression group was detected by the CIBERSORT algorithm. The results showed that both M1 and M2 macrophages accounted for higher proportions in the high UVRAG expression group, while the percentage of activated natural killer (NK) cells was higher in the low UVRAG expression group (Figure 4A). To find out the association between UVRAG expression and macrophage infiltration, GSEA was performed, and it was revealed that both immunoregulatory interactions between a lymphoid and a nonlymphoid cell and the chemokine signaling pathway were enriched in the high UVRAG expression group (Figure 4B). CC motif chemokine ligand 2 (CCL2), as the primary gene in the chemokine signaling pathway, exhibited a higher expression in the high UVRAG expression group (Figure 4C,D). In addition, UVRAG expression exhibited a positive correlation with infiltration of macrophages (r = 0.495) and a positive correlation with CCL2 expression (r = 0.402) (Figure 4E), indicating that UVRAG expression might influence infiltration of macrophages through CCL2. Given the role of UVRAG expression in genomic stability [17], PD-L1 expression was compared between the high and low UVRAG expression groups, and it was found that CD274 expression was higher in the high UVRAG expression group (Figure 4F). The expression of SP1, which has been identified as an upstream transcription factor to regulate CCL2 expression, was then examined. It was indicated that SP1 expression was upregulated when UVRAG was overexpressed, while it was downregulated when UVRAG was knocked down (Figure 3G). To identify that UVRAG upregulates CCL2 through SP1, we performed a knockdown experiment targeting SP1 in UVRAG overexpressed cell lines. The result showed that CCL2 expression was significantly downregulated when SP1 was knocked down in UVRAG overexpressed HCT116 (Figure 4G), suggesting that UVRAG may regulate CCL2 expression through SP1. Similarly, PD-L1 expression increased in UVRAG-overexpressed HCT116 cells and decreased in UVRAG-knockdown HT29 cells (Figure 3G).

### 3.7. Single-Cell Transcriptome Analysis Revealed the Underlying Mechanisms of Immune Modulation in the High UVRAG Expression Group

In the analysis of single-cell transcriptomic data from 6 patients with colon cancer (GSE188711), the cells were clustered into the following 13 clusters: B cells, CD4^+^ memory T cells, CD4^+^ naive T cells, CD8^+^ T cells, endothelial cells, epithelial cells, fibroblasts, macrophages, mast cells, monocytes, neutrophils, plasma, and regulatory T cells (Tregs) (Figure 5A). Comparison of UVRAG expression among these clusters indicated that UVRAG expression was slightly distinct among different cell types and the statistical difference was even significant (Figure 5E), demonstrating that UVRAG expression in RNA-seq bulk data was not influenced by the proportion of any specific cell type.

Epithelial cells with higher UVRAG expression showed higher expression of tumor migration-associated genes (HN1, HMGB2, PCNA, and CBX5) (Figure 5B). Then, the cell migration assay revealed that the migratory capability of UVRAG-overexpressed HCT116 cells was enhanced compared with the control group (Figure 5C). Furthermore, higher expressions of CXCL1, CXCL3, and CXCL8 in the high UVRAG expression group also indicated that UVRAG expression had a relationship with promoting tumor progression, angiogenesis, and immune suppression by recruiting tumor-associated macrophages (TMAs) [18,19,20]. Intensities of cell interactions between the high UVRAG expression group and the low UVRAG expression group were slightly discrepant (Figure 5D), and some distinct receptor–ligand pairs were found. The most prominent differences concentrated on MIF, CXCL, and interleukin pathways (Figure 5F), and tumor cells in the high UVRAG expression group tended to exert a stronger immunosuppressive effect by inhibiting immune cells and recruiting immunosuppressive cells. B and T lymphocyte attenuator (BTLA) in tumor cells could inhibit the activation of B cells, T cells, and macrophages by binding to TNFRSF14 [21]. The specific CCL15–CCR1 pair on tumor cells with high UVRAG expression and stromal cells could recruit tumor-associated neutrophils and CCR1^+^ myeloid cells, promoting tumor metastasis [22,23]. IL-1 from tumor cells with high UVRAG expression, involved in immunosuppression by interacting with several immune cells, mainly stimulated the recruitment and proliferation of myeloid-derived suppressive cells (MDSCs) [24]. IL-4 activated M2 macrophages by c-Jun N-terminal kinase-signal transducer and activator of transcription (JNK-STAT) pathway and impaired DC functions [25,26], indicating that higher expression of UVRAG in epithelial cells was correlated with tumor immune escape. Moreover, CXCL-16 inhibited the residency of CD8^+^ resident memory T cells (TRMs) and therefore facilitated tumor metastasis [27]. Ligand MIF, a protumor and antitumor mediator depending on tumor context, exhibited a stronger interaction with receptors on stromal cells in the high UVRAG expression group [28]. Collectively, tumor cells with high UVRAG expression had more intensive interactions with other cells and contributed to more immunosuppression and immune escape compared with tumor cells with low UVRAG expression.

## 4. Discussion

UVRAG has been generally accepted as a tumor suppressor in cancer [29], while in the present study, higher UVRAG expression was found to be negatively associated with the prognosis of CRC patients. Previous studies demonstrated that UVRAG plays a vital role in the process of autophagy [17], involving the clearance of senescent organelles and redundant proteins to keep intracellular homeostasis [30]. Autophagy, as a double-edged sword, exerts a suppressive effect on tumorigenesis because of its functions in limiting oxidative stress and enhancing antigen presentation [31,32]. On the other hand, with cancer progression, autophagy facilitates the survival of tumor cells under hypoxic stress and other harsh microenvironment [33]. The dual role of UVRAG may be related to the opposing role of autophagy in cancer.

In the present study, it was found that UVRAG expression had a negative correlation with CRC prognosis, and the potential mechanisms were associated with the effects of UVRAG on tumor malignancy and immunosuppression. The UVRAG expression could predict the survival of CRC patients in the subgroups of p53 mutation and CD44-positive, respectively. It is worth noting that the p53 mutant was reported to regulate apoptosis and autophagy by regulating Bax and Bcl2 [34], and UVRAG also had the same effect [4].

To further understand the role of UVRAG expression in tumor progression, the malignant behavior of tumor cells was explored, and higher UVRAG expression was identified as being associated with greater migration capacity. For the relationship between UVRAG expression and treatment resistance, it was found that UVRAG expression significantly affected the prognosis of patients undergoing adjuvant chemotherapy and was negatively correlated with the radiotherapy sensitivity in patients undergoing neoadjuvant radiotherapy. In vitro experiments also confirmed that UVRAG enhanced the resistance of CRC cells to radiotherapy and chemotherapy. A previous study demonstrated that UVRAG could contribute to resistance to chemotherapy and radiotherapy in an autophagy-dependent manner [35]. The increased UVRAG expression in HCT116-self and DCLK+ cells further indicated that UVRAG might be involved in regulating the stemness of CSCs. CD44 was reported as a marker of CRC stem cells. In the CD44-positive group, UVRAG expression was negatively correlated with OS. Autophagy has been shown to participate in supporting the generation of cells with high CD44 expression [36], which accounted for the discrepant survival results between CD44 high and CD44 low expression subgroups. A study reported that CSCs were associated with tumor chemotherapy/radiotherapy resistance [37]. CSCs have been confirmed in the majority of types of cancer [38,39,40]. During long-term conventional chemotherapy/radiotherapy, CSCs are mainly enriched in tumor tissue, resulting in therapeutic resistance [41].

The present study revealed that UVRAG expression was correlated with the expressions of PD-1, PD-L1, and PD-L2 in clinical data. In addition, PD-L1 expression was upregulated after overexpressing UVRAG. PD-1 and PD-L1 have been shown to affect autophagy, and a study showed that lower PD-1 expression could promote autophagy [42], while another study demonstrated that overexpression of PD-1/PD-L1 enhanced autophagy [43]. The present study suggested that UVRAG could regulate PD-L1 expression and subsequently influence tumor immunity. Therefore, it was supposed that UVRAG could not only affect the malignant behavior of tumor cells but also influence the immune microenvironment. A previous study revealed that UVRAG influenced peripheral naive T-cell homeostasis through an autophagy-independent pathway [44]. Moreover, UVRAG influenced the phagocytosis of macrophages by interacting with RUBCN [45]. The present study indicated that UVRAG could turn the tumor microenvironment to immunosuppression by secreting CCL2 to recruit macrophages, which was mediated by upregulating SP1 expression. It was also revealed that PD-L1 expression was upregulated when UVRAG expression was elevated, which might be another way for UVRAG to contribute to immune suppression. Analysis of scRNA transcriptomic data demonstrated that tumor cells with higher UVRAG expression might also influence macrophages through receptor–ligand pairs, such as BTLA–TNFRSF14 and CCL15–CCR1. Several studies have shown that the above-mentioned pathways could be potential candidates for antitumor therapy. TNFRSF14/BTLA, as an immune checkpoint, could be blocked to strengthen the activation of T cells [46]. Blockade of the CCL15–CCR1 axis could reverse the protumor inflammatory microenvironment by breaking recruitment of suppressive monocytes to inhibit tumor progression [47]. The IL-1β–IL1R axis, which only appeared in the interaction of tumor cells with the high UVRAG expression and myeloid cells, could be blocked to reverse immunosuppression and synergize with anti-PD-1 [48]. A series of autophagy-associated inhibitors has been developed to cope with chemoradiotherapy resistance [49]. UVRAG inhibition may restrain tumor progression by not only reducing tumor malignancy but also changing the immune environment. Furthermore, UVRAG inhibition could be more effective when combined with the inhibitor of the above-mentioned molecules, particularly the anti-PD-L1 antibody.

The present study shed new light on the potential of UVRAG as a novel therapeutic target. The association between UVRAG expression and tumor immunity was explored and its mechanism was clarified, providing evidence for UVRAG as a potential therapeutic target, while the efficacy of UVRAG inhibition and combined regimens should be assessed in future studies. In addition, whether the effect of UVRAG on tumor immunity depends on autophagy still needs further investigation.

## 5. Conclusions

In conclusion, it was revealed that high UVRAG expression was associated with poor survival, which might be mediated by the enhanced effect of UVRAG on radiotherapy resistance, chemotherapy resistance, tumor malignancy, and immunosuppression. In addition, UVRAG could recruit macrophages by upregulating SP1 in order to enhance CCL2 expression and regulate PD-L1 expression. This study advanced the understanding of UVRAG from the aspect of tumor immunity and confirmed the potential of UVRAG as a novel therapeutic target.

## Figures and Tables

**Figure 1 cancers-15-02502-f001:**
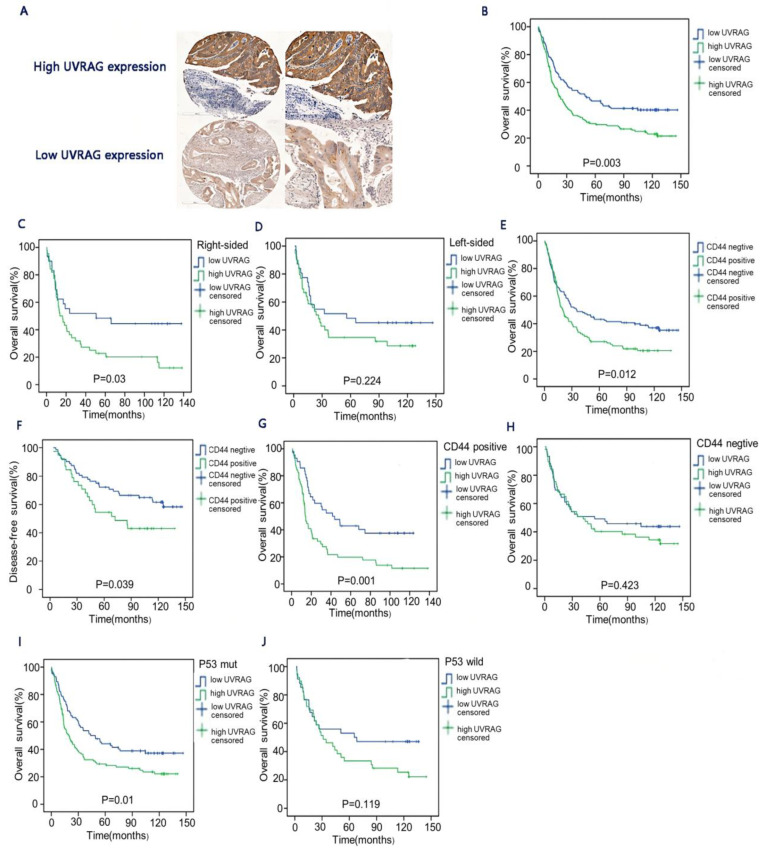
UVRAG expression and prognosis in colorectal cancer. (**A**) Representatives of IHC staining for high UVRAG expression and low UVRAG expression. (**B**) Overall survival in the high UVRAG expression group and the low UVRAG expression group (*n* = 294; *p* = 0.003). (**C**) Overall survival in the high UVRAG expression group and the low UVRAG expression group in right-sided colon cancer (*n* = 177; *p* = 0.03). (**D**) Overall survival in the high UVRAG expression group and the low UVRAG expression group in left-sided colon cancer (*n* = 117; *p* = 0.224). (**E**) Overall survival of patients with CD44-positive and CD44-negative (*n* = 235; *p* = 0.012). (**F**) Disease-free survival of patients with CD44-positive and CD44-negative (*n* = 235; *p* = 0.039). (**G**) Overall survival in the high UVRAG expression group with CD44-positive and the low UVRAG expression group with CD44-positive (*n* = 109; *p* = 0.001). (**H**) Overall survival in the high UVRAG expression group with CD44-negative and the low UVRAG expression group with CD44-negative (*n* = 126; *p* = 0.423). (**I**) Overall survival in the high UVRAG expression group with p53 mutation and the low UVRAG expression group with p53 mutation (*n* = 155; *p* = 0.01). (**J**) Overall survival in the high UVRAG expression group with wild-type p53 and the low UVRAG expression group with wild-type p53 (*n* = 74; *p* = 0.119). All patients were from Cohort 1. All the *p* values of survival analysis were calculated by log-rank test.

**Figure 2 cancers-15-02502-f002:**
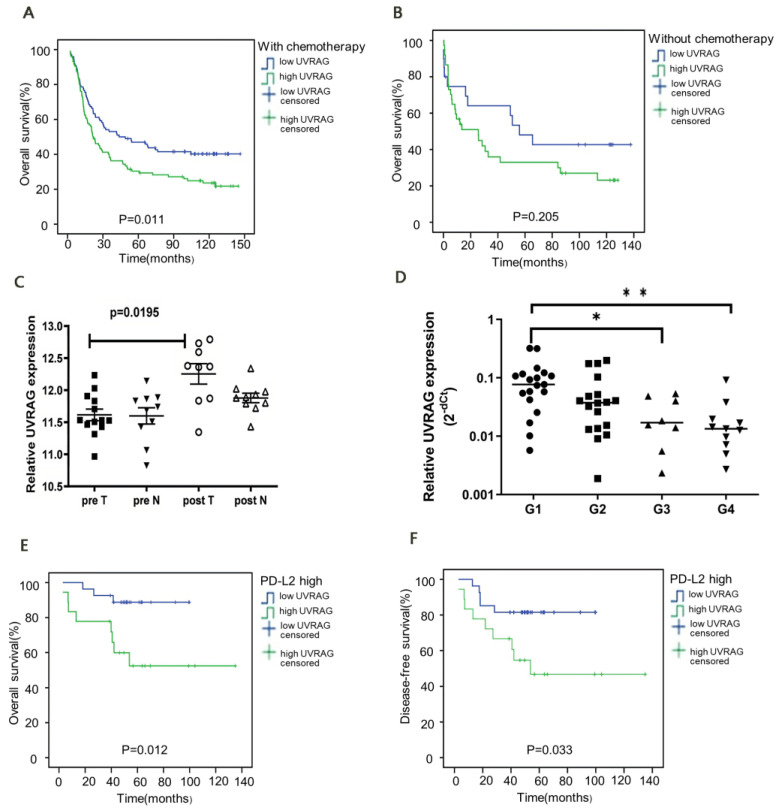
UVRAG expression was correlated with the efficacy of chemoradiotherapy and prognosis. (**A**) Overall survival in the high UVRAG expression group and the low UVRAG expression group after receiving adjuvant therapy (*n* = 228; *p* = 0.011). (**B**) Overall survival in the high UVRAG expression group and the low UVRAG expression group without receiving adjuvant therapy (*n* = 63; *p* = 0.205). (**C**) Relative expression of UVRAG in tumor and normal samples before and after radiotherapy (*p* = 0.0195). (**D**) Relative expression of UVRAG in different TRG grades. (**E**) Overall survival in the high UVRAG expression group with the higher PD-L2 expression and in the low UVRAG expression group with the higher PD-L2 expression (*n* = 46; *p* = 0.012). (**F**) Disease-free survival in the high UVRAG expression group with the higher PD-L2 expression and in the low UVRAG expression group with the higher PD-L2 expression (*n* = 46; *p* = 0.033). Independent-samples *t*-test (**C**,**D**). Patients were from Cohort 1 (**A**,**B**). Patients were from Cohort 2 (**E**,**F**). Patients were from Cohort 3 (**D**). All the *p* values of survival analysis were calculated by log-rank test. * *p* < 0.05, ** *p* < 0.01.

**Figure 3 cancers-15-02502-f003:**
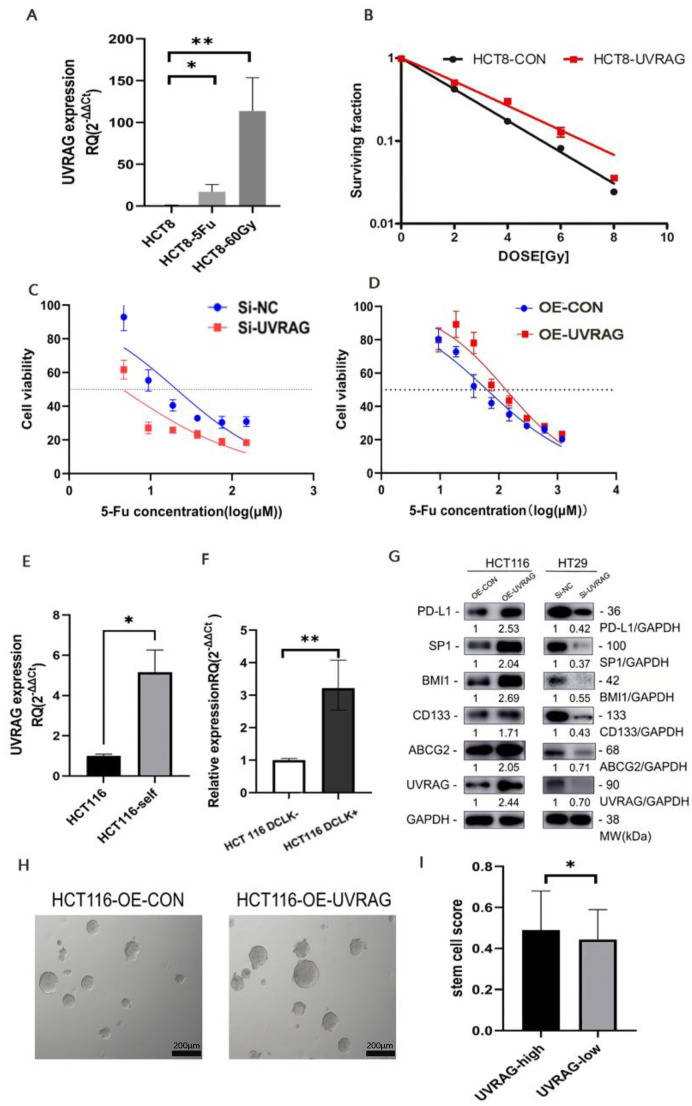
Correlation between UVRAG expression and therapeutic resistance. (**A**) Relative expression of UVRAG in HCT8, HCT8-5Fu, and HCT8-60Gy cells. (**B**) The sensitivity of HCT8-CON and HCT8-UVRAG cells to radiotherapy. (**C**) UVRAG knockdown increased 5-Fu sensitivity. (**D**) UVRAG overexpression decreased 5-Fu sensitivity. (**E**) Relative mRNA expression of UVRAG in HCT116 and HCT116-self cells. (**F**) Relative mRNA expression of UVRAG in HCT116 DCLK- and HCT116 DCLK+ cells. (**G**) Western blot of PD-L1, UVARG, ABCG2, BMI1, and SP1 after upregulating UVARG or downregulating UVRAG. The uncropped blots are shown in Appendix A. (**H**) Spheroid formation in UVRAG overexpressed HCT116 and its control cells. (**I**) Stem cell score evaluated via ssGSEA in patients with UVRAG higher expression and UVARG lower expression. Independent-samples *t*-test (**A**,**E**,**F**). Mann Whitney U test (**I**). * *p* < 0.05, ** *p* < 0.01.

**Figure 4 cancers-15-02502-f004:**
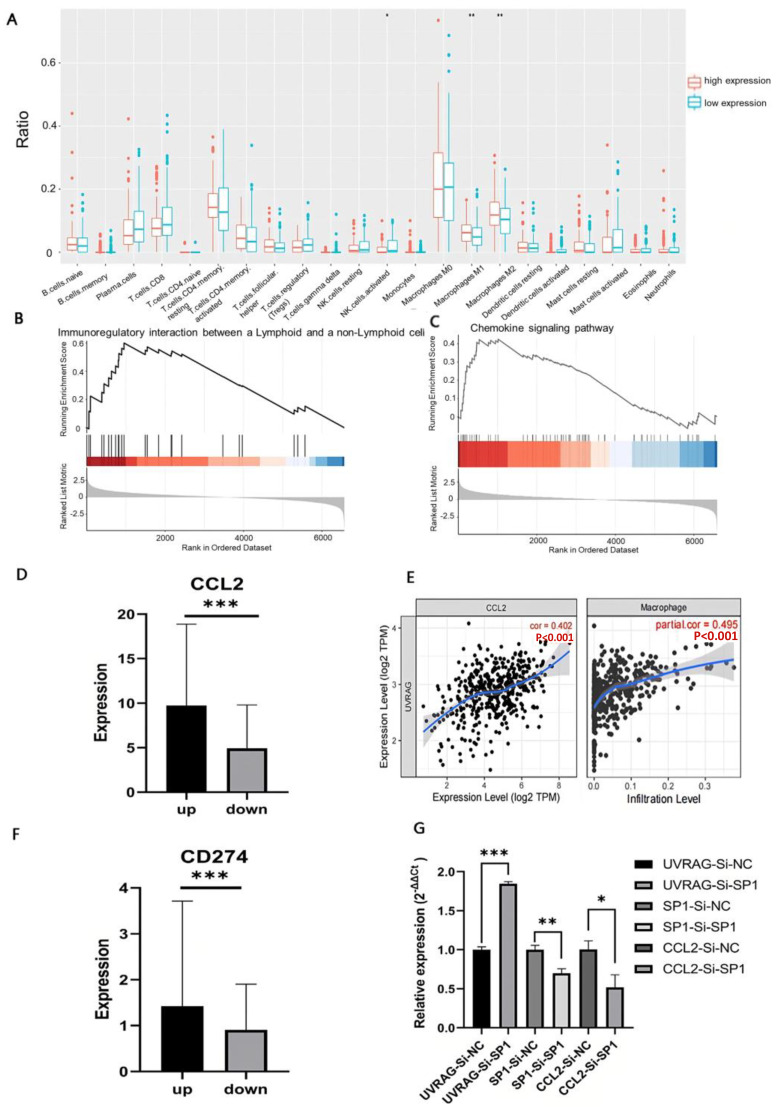
Correlation between UVRAG expression and tumor immunity. (**A**) Proportion of immune cells in the high UVRAG expression group and low UVRAG expression group. (**B**) Enrichment of immunoregulatory interactions between lymphoid and nonlymphoid cell pathways. (**C**) Enrichment of chemokine signaling pathway. (**D**) CCL2 expression in the high UVRAG expression group and the low UVRAG expression group. (**E**) Correlation between UVRAG expression and the infiltration of macrophages. (**F**) CD274 expression in the high UVRAG expression group and the low UVRAG expression group and CCL2 expression. (**G**) Relative mRNA expression of UVARG, SP1, and CCL2 in UVRAG overexpressed HCT116 with SP1 knocked down. Independent-samples *t*-test (**A**,**D**,**F**,**G**). * *p* < 0.05, ** *p* < 0.01, *** *p* < 0.001.

**Figure 5 cancers-15-02502-f005:**
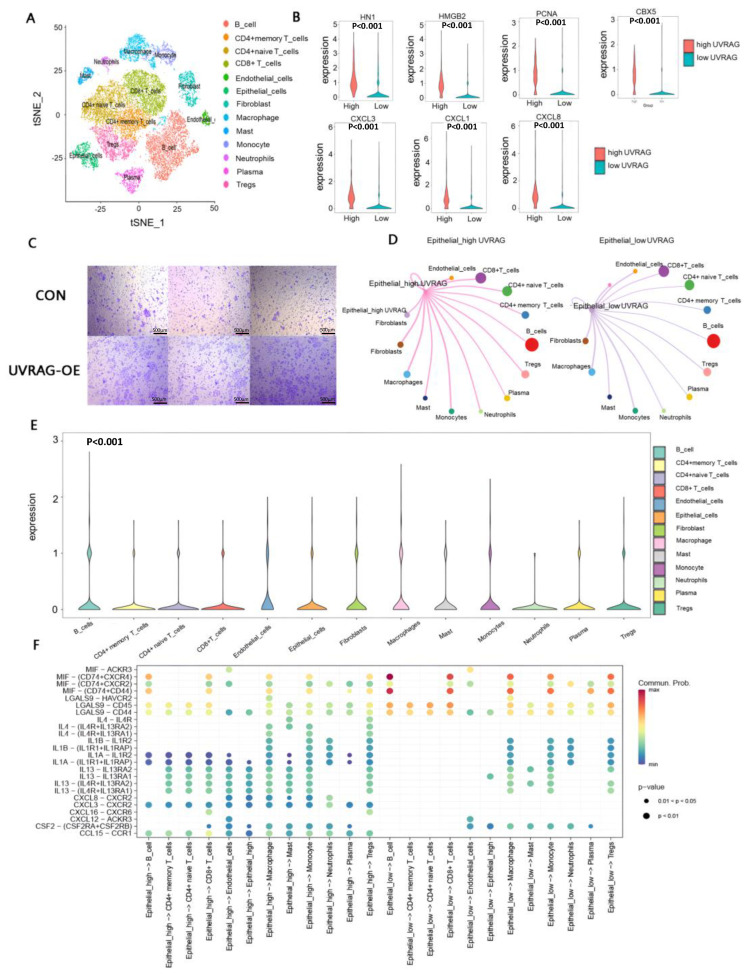
Analysis of scRNA-seq data. (**A**) t-SNE plot of 21,579 cells from 6 patients with CRC. (**B**) Violin plots of the expressions (log2(TMP + 1)) of PCNA, HMGB2, HN1, CBX5, CXCL3, CXC1, and CXCL8. (**C**) Migratory capability was enhanced after UVRAG overexpression. (**D**) Circle plot of communication of tumor cells with high UVRAG expression to other stromal cells. (**E**) UVRAG expression in different cell types. (**F**) Bubble plot of ligand–receptor pairs of tumor cells to stromal cells. Independent-samples *t*-test (**B**). One-way ANOVA (**E**).

**Table 1 cancers-15-02502-t001:** Correlation between UVRAG and clinical features.

Clinical Pathological Data	UVRAG
differentiation	
R	0.153
*p* value	0.016
T stage	
r	0.047
*p* value	0.445
N stage	
r	0.056
*p* value	0.367
M stage	
r	0.246
*p* value	<0.0001
TNM stage	
r	0.163
*p* value	0.008
Histological type	
U	0.6687
*p* value	0.5369
Vascular cancer embolus	
U	1.57
*p* value	0.1155
Location (colon & rectal)	
U	1.45
*p* value	0.14
PD-1	
r	0.128
*p* value	0.041
PD-L1	
r	0.236
*p* value	0.000
PD-L2	
r	0.278
*p* value	0.000

r for Pearson correlation analysis; U for Kendall correlation analysis.

**Table 2 cancers-15-02502-t002:** Correlation between location and clinical features.

Clinical Pathological Data	Location (Colon & Rectal)
UVRAG	
U	1.45
*p* value	0.14
PD-L2	
U	1.99
*p* value	0.046
PD-L1	
U	1.78
*p* value	0.074
PD-1	
U	0.78
*p* value	0.43

r for Pearson correlation analysis; U for Kendall correlation analysis.

**Table 3 cancers-15-02502-t003:** Correlation between UVRAG and clinical features in colon cancer.

Clinical Pathological Data	UVARG (Colon)
PD-L2	
r	0.319
*p* value	0.002
PD-L1	
r	0.315
*p* value	0.000
PD-1	
r	0.332
*p* value	0.001

r for Pearson correlation analysis; U for Kendall correlation analysis.

**Table 4 cancers-15-02502-t004:** Correlation between UVRAG and clinical features in rectal cancer.

Clinical Pathological Data	UVRAG (Rectal)
PD-L2	
r	0.232
*p* value	0.004
PD-L1	
r	0.124
*p* value	0.12
PD-1	
r	−0.023
*p* value	0.78

r for Pearson correlation analysis; U for Kendall correlation analysis.

**Table 5 cancers-15-02502-t005:** Correlation between UVRAG and immune checkpoint.

Clinical Pathological Data	UVRAG
PD-L2	
r	0.222
*p* value	0.001
PD-L1	
r	0.228
*p* value	0.000

## Data Availability

The data presented in this study are available in this article and the Appendix A.

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
