# Peer review of "UVRAG Promotes Tumor Progression through Regulating SP1 in Colorectal Cancer"

_cancers, 2023, doi:10.3390/cancers15092502_

Round 1

Reviewer 1 Report (Previous Reviewer 1)

This study provides a thorough investigation of the role of UVRAG in CRC. Through a combination of immunohistochemistry, RNA sequencing (RNA-seq), and in vitro experiments, the authors explore the relationship between UVRAG and CRC prognosis, treatment response, and underlying mechanisms. While some concerns were raised, the authors have made commendable efforts to address them, and the experimental findings provide support for the authors' claims, highlighting the potential of UVRAG as a therapeutic target in CRC. Overall, this study contributes to a better understanding of UVRAG's role in CRC and provides valuable insights for future investigations in this field.

Reviewer 2 Report (Previous Reviewer 2)

I am satisifed by the corrections made by the authors

This manuscript is a resubmission of an earlier submission. The following is a list of the peer review reports and author responses from that submission.

Round 1

Reviewer 1 Report

The authors describe the role of UVRAG in colon cancer progression and prognosis. The authors have done a good job at displaying the importance of UVRAG by utilizing colon cancer patient samples and in vitro analysis. Overall, the paper seems comprehensive, though further exploration on the role of UVRAG on the colon cancer stem cell population would improve the implication of this work.

Here are some of the comments/suggestions that the authors could elaborate on

1.     Were other colon stem cell markers like LGR5, CD133, ALDH1 and EpCam also looked at to understand its relationship with UVRAG expression?

2.     An important property of cancer stem cells in self-renewal. Was any self-renewal marker like c-Myc or BMI-1 also looked at?

3.     What is the proportion of Cancer stem cells in UVRAG (high expression) vs low expression tissues? This could provide a better idea of the role of UVRAG in cancer stem cell induced tumor progression.

4.     Can the authors provide a rationale for increase UVRAG expression in patients receiving adjuvant radiotherapy?

5.     The authors utilize stem cell enriched spheres to detect the expression of UVRAG in cancer stem cells. However, the effect of increased UVRAG on the self-renewal ability of these cells would provide more comprehensive evidence of the role of UVRAG on stem cell enrichment. Comparison of spheroid growth in UVRAG over expressing or UVRAG knockdown cells vs wild-type cells could help prove this.

The authors show increased PDL1 expression in UVRAG-overexpressed cells and vice a versa in UVRAG-knocked down cells. Were cancer stem cell/self-renewal markers also looked at in these cells? The results of these experiments would help identify the relationship between UVRAG expression and cancer stem cells.

Author Response

Thanks for your kind advice. The responses to your suggestion are as follows:

Comment 1: Were other colon stem cell markers like LGR5, CD133, ALDH1, and EpCam also looked at to understand its relationship with UVRAG expression?

Response: We conducted a Western blot to explore the relationship between colon stem cell markers and UVRAG expression, and we found that the expression of colon stem cell markers including CD133, ABCG2, and BMI-1 was upregulated when UVRAG was overexpressed. Similarly, the expression of CD133, ABCG2, and BMI-1 was downregulated when UVRAG was knockdowned. We have revised the content on page 13, specifically lines 341-347, and made corresponding changes in Figure 3G and its legend, which have been highlighted in yellow.

Comment 2: An important property of cancer stem cells in self-renewal. Was any self-renewal marker like c-Myc or BMI-1 also looked at?

Response: We explored the BMI-1 expression and found that it is upregulated in UVRAG overexpressed cells, and downregulated in UVRAG knockdowned cells. We have revised the content on page 13, specifically lines 341-347, and made corresponding changes in Figure 3G and its legend, which have been highlighted in yellow.

Comment 3: What is the proportion of Cancer stem cells in UVRAG (high expression) vs low expression tissues? This could provide a better idea of the role of UVRAG in cancer stem cell induced tumor progression.

Response: To explore the proportion of cancer stem cells in tissues with high UVRAG expression and tissues with low UVRAG expression, we use ssGSEA (single sample Gene Set Enrichment Analysis) algorithm to assess the cancer stem cell score of RNA-seq data from TCGA, which is acquired from 240 patients with colon cancer. We found that cancer stem cell score of UVRAG high expression group was significantly higher than that in UVRAG low expression group (P=0.03). We have revised the content on page 13, specifically lines 351-354, and made corresponding changes in Figure 3I and its legend, which have been highlighted in yellow.

Comment 4: Can the authors provide a rationale for increase UVRAG expression in patients receiving adjuvant radiotherapy?

Response:  In our study, the expression of UVRAG in rectal cancer tissue was negatively correlated with the efficacy of radiotherapy. Patients with higher expression of UVRAG in pre-treatment biopsies had poor TRG grading. In vitro experiments have confirmed that the high expression of UVRAG in CRC cells promoted the radiation resistance of tumor cells, and the expression of UVRAG in radioresistant cells was higher than in control cells. These results indicated that UVRAG was associated with radio-resistance. UVRAG was highly expressed in DCLK1 positive cancer cells and HCT116-self cells. UVRAG promoted the expression of stem cell markers in cancer cells and enhanced the spheroid formation ability of CRC cells. These results suggested that UVRAG could promote the stemness of tumor cells.  Cancer stem cells were reported to be associated with radiotherapy resistance. Ionizing radiation induces double-strand breaks in genomic DNA in cancer cells, leading to genomic instability, cell cycle checkpoint arrest, apoptosis, or death. Tumor cells may evolve personalized DNA damage responses (DDRS) to combat IR and avoid being killed. Cancer stem cells activate CHK1 and CHK2 more efficiently than parental contributing to enhance DNA damage repair and radioresistance. (Petroni G, et al. Radiotherapy as a tool to elicit clinically actionable signaling pathways in cancer. Nat Rev Clin Oncol. 2022 Feb;19(2):114-131). Previous study reported that radiotherapy activated stemness pathway and enriched cancer stem cells with higher resistance to radiotherapy (Ghisolfi L, Keates AC, Hu X, Lee DK, Li CJ. Ionizing radiation induces stemness in cancer cells. PLoS One. 2012;7(8):e43628). We think that after receiving radiotherapy, cancer stem cells that are resistant to radiotherapy will be enriched, resulting in an enrichment of high UVRAG expression cells.

Comment 5: The authors utilize stem cell enriched spheres to detect the expression of UVRAG in cancer stem cells. However, the effect of increased UVRAG on the self-renewal ability of these cells would provide more comprehensive evidence of the role of UVRAG on stem cell enrichment. Comparison of spheroid growth in UVRAG over expressing or UVRAG knockdown cells vs wild-type cells could help prove this.

Response: We compared the spheroid growth in UVRAG overexpressed cells and that in control cells, and we found that UVRAG overexpressed cells had enhanced self-renewal ability when compared to the control. We have revised the content on page 13, specifically lines 347-350, and made corresponding changes in Figure 3H and its legend, which have been highlighted in yellow.

If there is any other problem, please don’t hesitate to contact us. We are willing to listen to your advice and make it better.

Reviewer 2 Report

This paper investigates the role of the UVRAG gene in cancer progression.

The major part of the paper is devoted to the analysis of correlations between UVRAG expression and various characteristics of colorectal carcinomas. I suggest to limit the paper by this fragment of the study. Experiments with cell lines require more thorough description, with an utmost attention to experimental design, composition of controls, validation of the observed effects, etc.  – they may deserve an independent publication if properly presented.

How the authors classify the tumors for high vs. low expressors?

Please reduce the number of tables. For example, Tables 1-3 may be presented as a single table, and I recommend to move this information to the Supplement.

English is understandable bot does achieve the level necessary for a publication in scientific journal. Please seek for someone who is proficient in English biomedical writing.

Author Response

Thanks for your kind advice. The responses to your suggestion are as follows:

Comment 1: Experiments with cell lines require more thorough description, with an utmost attention to experimental design, composition of controls, validation of the observed effects, etc.  

Response: We added more details about our experiments, and we are going to further explore the deeper mechanism in future research. We have revised the content on page 12, specifically lines 322-323, 325-326, and 333-335, which have been highlighted in yellow for easy reference.

Comment 2: How the authors classify the tumors for high vs. low expressors?

Response: The expression of UVRAG was determined by assessing the intensity of staining, as well as the percentage of staining cells. Specifically, tissues that showed no staining were assigned a score of 0. Tissues that showed a staining with less than 25% of cells were scored as 1. Tissues with strong staining observed in 25-50% of cells were scored as 2, while those with strong staining in more than 50% of cells were given a score of 3. The patients with low UVRAG expression refer to patients who were scored as 0 or 1, and the patients with high UVRAG expression refer to patients who were scored as 2 or 3. We have revised the content on page 5, specifically lines 106-113, which have been highlighted in yellow for easy reference.

Comment 3: Please reduce the number of tables.

Response: According to your advice, we moved Tables 1-3 to supplemental information.

Comment 4: English is understandable bot does achieve the level necessary for a publication in scientific journal. Please seek for someone who is proficient in English biomedical writing.

Response: According to your advice, we resorted to a professional polishing agency for proofreading our manuscript.

If there is any other problem, please don’t hesitate to contact us. We are willing to listen to your advice and make it better.